# Smart Bio-Impedance-Based Sensor for Guiding Standard Needle Insertion

**DOI:** 10.3390/s22020665

**Published:** 2022-01-15

**Authors:** Ivan Kudashov, Sergey Shchukin, Mugeb Al-harosh, Andrew Shcherbachev

**Affiliations:** Department of Medical and Technical Information Technology, Bauman Moscow State Technical University, 105005 Moscow, Russia; schookin@bmstu.ru (S.S.); alharosh@bmstu.ru (M.A.-h.); Shcherbachev_av@bmstu.ru (A.S.)

**Keywords:** puncture identification, electrode system, electrical impedance

## Abstract

A venipuncture is the most common non-invasive medical procedure, and is frequently used with patients; however, a high probability of post-injection complications accompanies intravenous injection. The most common complication is a hematoma, which is associated with puncture of the uppermost and lowermost walls. To simplify and reduce complications of the venipuncture procedure, and as well as automation of this process, a device that can provide information of the needle tip position into patient’s tissues needs to be developed. This paper presents a peripheral vascular puncture control system based on electrical impedance measurements. A special electrode system was designed to achieve the maximum sensitivity for puncture identification using a traditional needle, which is usually used in clinical practice. An experimental study on subjects showed that the electrical impedance signal changed significantly once the standard needle entered the blood vessel. On basis of theoretical and experimental studies, a decision rule of puncture identification based on the analysis of amplitude-time parameters of experimental signals was proposed. The proposed method was tested on 15 test and 9 control samples, with the results showing that 97% accuracy was obtained.

## 1. Introduction

The venipuncture is a common procedure in clinical practice for medical conditions such as requiring blood draw and drug and fluid administration, which can be performed by the use of needles and catheters within the lumen of a vein [1]. For the correct execution of these manipulations, an accurate understanding of the position of the injection needle relative to the blood vessel is necessary, otherwise post-injection complications associated with both walls of the vessel being punctured or partial penetration of the needle into the vein lumen might occur. Nowadays, this procedure is mainly based on the surgeon’s experience. Automation of this process, based on objective information of the needle tip position, obtained by instrumental methods, can reduce the post-injection complications and increases the success rate of peripheral venous cannulation. Several methods and systems have been developed for guiding the needle insertion. The use of ultrasound guidance for peripheral intravenous has become standard practice; however, this method is associated with the involvement of complex, expensive equipment and a specialist in the ultrasound diagnostics system [2]. Recently, optical methods have been widely used for peripheral vein visualization; however, the optical-based methods cannot allow controlling the penetration of the injection needle into the lumen of the vein [3]. The most popular methods are those based on sensing. However, this method is very sensitive to motion artifacts, which cannot be avoided during the venipuncture procedure. Several studies have been carried out to determine the tissue types based on the difference in electrical conductivity of biological tissues through which the needle electrode moves, such as skin tissue, connective and muscle tissues, bone, and blood [4]. These methods use a special needle-electrode with a conductive end and an insulating base, which limits their use for daily procedures such as venipuncture. In this paper, we present a novel bio-impedance sensor for guiding traditional needle insertion, which is used in daily medical practice and absolutely does not require special needle manufacturing. To use the bio-impedance guidance, a biotechnical system was considered, which is physically based on a biological object—an area of the forearm, consisting of a complex of biological tissues.

The tissues in the region of interest have different electrical properties, as illustrated in Table 1. Thus, an electrical current with 100 kHz was considered in this work as the impact of the capacitive part, which was determined by the heterogeneity of tissue structures in the region of interest to be less than 10% [5].

Moreover, as the specific electrical resistance of venous blood is several times lower than that of the surrounding tissues [6], it is in principle possible to determine the moment of needle-electrode penetration into the blood [7,8,9,10,11,12,13,14,15,16] because, in this case, the measured impedance of the needle relative to the reference electrode decreases. Figure 1 shows a simplified diagram of the proposed measuring scheme, with the sensor based on sending a small current through the needle for electrical impedance recoding in a two-electrode setup. An alternating current of frequency 100 kHz and current force up to 1 mA were used to avoid nerve and muscle stimulation [17,18,19,20,21,22,23,24,25,26].

## 2. Materials and Methods

### 2.1. Numerical Modeling

To perform the desired electrical impedance measurements, a special electrode system should be designed to achieve the maximum sensitivity for puncture identification [27]. However, to substantiate the adequate electrode system position as well as the optimal parameters of the electrode system, such as the distance between needle and attached electrode, as well as the contact area size of the attached electrode, it is necessary to understand the current line distribution between the electrodes; hence, numerical modeling using SEMCAD X 14.8 (SPEAG AG, Zurich, Switzerland) was proposed in this study.

To mimic the study area, a 3D model was developed that consists of two spaces. The first space is a homogeneous space and has the electrical resistivity of muscle tissues, which is equal to 5 Ohm·m [28], while the second space is the simulated venous vessel with electrical resistivity close to the blood resistivity and equal to 1.5 Ohm·m [28]. The geometrical size of the designed model with vessel was selected according to the real size of the forearm as well as the possible location and depth of venous blood vessels. The overall dimensions of the model shown in Figure 2 are 200 × 150 × 50 mm, while the vessel has a 5 mm diameter and is located at a 5 mm depth. To simulate the electromagnetic field and study the current distribution, an electrode system was added to the model. The electrode system comprises two electrodes; the first electrode was attached on the surface of the model directly above the vessel, while the second electrode is the needle itself, and the overall parameters of the needle electrode correspond to the size of a standard injection needle of G21 caliber: length 40 mm and diameter 0.8 mm.

### 2.2. Experimental Setup

The experimental studies on subjects were carried out in the medical and technological center of Bauman Moscow State technical university, and research ethics was followed. All subjects were informed about the whole procedure as well as the mythology before experiments, and then provided written informed consent. Five human subjects were involved during this study and 15 venipunctures were performed for the five subjects. The corresponding electrical impedance measurement was performed using the ReoCardioMonitor system, which was developed at the Department of Medical and Technical Information Technology at Bauman Moscow State technical university and has certification documents for clinical trials [29]. The RheoCardioMonitor allows real-time recording of electrical impedance on a personal computer and provides a four-electrode technique setup; the technical specification of this system is illustrated in Table 2. The bio-sensor proposed in this study requires electrical impedance recoding in a two-electrode setup, thus the RheoCardioMonitor system was adapted to provide the bio-impedance recording with this type of electrode configuration by combining the current and measuring channels.

### 2.3. The Puncture Identification Algorithm

This work provides a solution for the daily routine procedure of venipuncture, thus several mechanisms accompany manual needle insertion, which can mask the detection of internal puncture. In order to decrease the effect of these mechanisms, a smart algorithm that allows accurate detection should be developed. However, the procedure of venipuncture using a needle-electrode with some assumption can be considered as a procedure of current flowing through the earth using a ground rod, hence the mathematical model presented in Equation (1) was used to study the influence of mechanisms that can accompany manual insertion.
(1)$$Z=\frac{\rho }{2\pi l}ln\frac{4x}{d_{0}}$$
where *Z*—the electrical impedance changes, ρ—the apparent electrical resistivity of tissues, *x*—the depth of needle electrode penetration, and $d_{0}$—the needle diameter.

However, the normalization of the experimental signal that can be obtained from subjects by the proposed mathematical model can allow the detection of venipuncture as a linear dependence should be observed when the needle moves in soft tissues; once the needle penetrates the vessel, the electrical impedance signal will change the trajectory, in contrast to the mathematical model approximation. According to this criterion, it will be possible to express a puncture of the vein wall. However, in this case, the experimental signal should be obtained with some assumptions. During the electrical impedance recording, the rate of needle insertion should be uniform and the patient should not perform any movement. However, these assumptions cannot be realized because, when performing venipuncture in real clinical conditions, it is impossible to achieve from medical personnel a uniform rate of needle insertion during the procedure of venipuncture without special equipment. Therefore, it is necessary to take into account the speed of needle insertion and the depth of its penetration into soft tissues during the normalizing of the experimental recording; consequently, Equation (1) was presented in the following form:(2)$$\frac{dz}{dt}=\frac{dz}{dx}V\left(t\right)$$
where *V*(*t*)—the rate of needle electrode motion in tissues and dz/dx—the spatial sensitivity function.

The proposed mathematical model shows that the dz/dt dependence has two components: the needle-electrode speed and the spatial sensitivity function. The needle-electrode speed depends on human factors, while the detail analysis of spatial sensitivity function can determine the normalization factor to reduce the motion artifacts. Thus, according to the mentioned concepts, the spatial sensitivity function can be characterized by Equation (3).
(3)$$\frac{dz}{dx}=\frac{2\pi }{\rho d_{0}}\left(\frac{1-ln\;\frac{4x}{d}}{ln^{2}\;\frac{4x}{d}}\right)\cdot Z^{2}$$

The function before $Z^{2}$ is a relatively small value and can be neglected. Thus, if dz/dt is normalized to $Z^{2}$, the impedance changes will mainly be determined by the puncture of the vessel wall and uneven speed, and not by the depth of the blood vessel, which in turn will improve the accuracy of determining the puncture of the vein wall.

## 3. Results

### 3.1. The Result of Numerical Modeling

Figure 3 shows the results of numerical modelling, which was conducted using SEMCAD X 14.8 to select the appropriate specification of the electrode system as well as the right placement of the attached electrode. In Figure 3a, several needle positions were considered during the modeling, thus the needle was removed from the attached electrode by 20, 40, 60, 80, and 100 mm. For every location, the needle-electrode was submerged by 3, 7.5, and 13 mm. The 7.5 mm insertion depth corresponds to the first puncture of the vessel wall, whereas the 13 mm insertion depth of needle electrode corresponds to a double puncture of a venous vessel, when the needle penetrates the uppermost and lowermost walls. For each needle position, the relative change in electrical impedance due to puncture was estimated by Equation (4).
(4)$$\delta =\frac{Zmax-Zmin}{Zmax}*100\%$$
where *Z_max_*—the electrical impedance value for needle-electrode insertion to a depth of 3 mm, and *Z_min_*—the electrical impedance value for needle-electrode insertion to a depth of 7.5 mm.

The needle insertion depths at 3 mm and 7.5 mm were considered as a criterion for assessing the relative change in impedance because, at these depths, the needle intersects the plane of the second medium, which is a simulation of the first puncture of the blood vessel wall. As shown in Figure 3a, the smaller the distance between the attached electrode and the needle, the larger the relative change in electrical impedance to puncture. As a result, a minimum distance of 20 mm was set for further experimental studies.

The results of the optimal selection of attached contact area size are shown in Figure 3b. During this modelling, the distance between the attached electrode and the needle was fixed to 20 mm, while the contact area of the attached electrode was changed by this order 1 × 1, 3 × 3, 5 × 5, 10 × 10, 20 × 20, and 30 × 30 mm. The attached electrode was installed above the imitation vessel on the surface of the model and, correspondingly, the needle electrode was inserted into 3 and 7.5 mm depths. As shown in Figure 3b, when the contact area reached 20 × 20 mm, the values of the relative change in electrical impedance began to approach the maximum value. Thus, the optimal contact area should be at least 20 × 20 mm for the best visualization of the moment of the first puncture of the blood vessel wall during venipuncture in the forearm region on the peripheral veins.

As a result of numerical modeling, the electrode system shown in Figure 4 was designed for experimental studies on subjects. The electrode system is attached to the skin current electrode, the contact area of the attached electrode is 20 × 30 mm, and it is made of medical-grade stainless steel. A special clamp was developed to make a traditional standard injection needle work as a needle electrode without breaking the rules of asepsis. The clamp is a bent rod fastener; a compression spring is attached to the rod, which is placed in a rectangular plastic case. Pressing on the end part of the body from the opposite side causes a curved rod to extend and the injection needle can be fixed. After fixing the injection needle, the end is released and the spring is released. The curved rod rushes into the body and grips the injection needle. The rod is made of AISI 304 stainless steel. The overall dimensions of the clamp are 40 × 5 mm. The size of the area for needle fixation is 2 mm. However, the electrode system based on a bio polar setup, and the same pair of electrodes is used both for excitation and measurement. Thus, the electrode system is attached to the skin electrode shown in Figure 4a with a dimension greater than the vessel diameter, and should be placed over the vein from where the blood is drawn, while the second electrode is the clamp, which should be connected to the needle to make it work as an electrode, as shown in Figure 4b.

### 3.2. The Results of the Experimental Study

The experimental setup is shown in Figure 5. The cephalic vein was selected because this vein is peripheral, easily available, and commonly used for blood collection [30]. The attached electrode system was positioned around the vein from which the blood was drawn in the arm area above the needle insertion location. The needle was inserted at about 10–20° degrees to the skin and moved towards the attached electrode. However, the angle of needle insertion did not affect the bio-impedance sensing for the venous entry detection during venipuncture [31,32,33,34,35].

Figure 6 shows the obtained electrical impedance change and it first derivative. The electrical impedance value decreases in correspondence with the rate of needle electrode injection; once the needle entered the blood vessel, the signal changed significantly. The analysis of the electrical impedance signal obtained during experimental studies showed that it is advisable to identify a puncture using a differential signal because, at the moment of venous vessel puncture, a jump occurs, indicating a transition from a less conductive medium to a more conductive medium. To study the differential signal, the received signal was transformed by Equation (5), which provides the smoothed derivative.
(5)$$H\left(Z\right)=\frac{1}{10T}\left(-2Z^{-2}-Z^{-1}+Z^{1}+2Z^{2}\right)$$
where *T*—the sampling period and *Z*—the recoding bio-impedance signal.

As shown from the corresponding first signal derivative, the similar complexes of puncture impedance changes can cause the puncture to be masked and hard to detect. However, the detailed analysis of the impedance signal and its derivative shows that, in the process of needle-electrode penetration into soft tissues, complexes of impedance changes similar to a puncture can appear that mask the main desired event and make it hard to identify. Thus, for the obtained experimental samples, consisting of fifteen experimental signals, the normalization procedure proposed in Section 2.3 was applied to decrease the influence of the processes affecting the electrical impedance changes associated with the puncture event. As shown in Figure 7, the applied normalization procedure could increase the signal to noise ratio (SNR) to an average of 24 dB, which helps to determine the moment of the first puncture of the blood vessel wall.

The collected data from all subjects were analyzed and the corresponding amplitude-time characteristics are illustrated in Table 3. As shown in Table 3 for every signal, the following amplitude-time characteristics were determined: ΔZcommon—the measured impedance range; ΔZpuncture—the impedance change during vein wall puncture; Tpuncture—the time of puncture of the vein wall.

## 4. Discussion

To propose an algorithm of puncture identification, the function *X*1 in Equation (6), which describes the normalization of the signal first derivative by the $Z^{2}$, was analyzed for such cases like puncturing and in the absence of puncture.
(6)$$X1=\frac{dz}{dt}\frac{d_{0}}{Z^{2}}$$

During this study, it was noted that the amplitude-time characteristics of *X*1 function at the puncture moment and at artifact moment were absolutely different. For more detailed descriptions of these events, it was proposed to conduct a function contour analysis, which allows developing an algorithm for puncture identification. Figure 8 shows the change of *X*1 function (complex—a puncture candidate). In Figure 8, the signal was inverted and multiplied by (–) for simplification as well as to get rid of the minus before the function in Figure 7b. To determine the boundaries of events, the *X*1 function was integrated and differentiated. The analysis of the integral and differential signals made it possible to determine the time intervals between the reference points of the function, which characterize the event under study. Thus, the obtained parameters of *X* function are the result of needle electrode interaction with the venous vessel wall during venipuncture and the needle electrode insertion into soft tissue.

According to the proposed algorithm and the contour analysis, the following amplitude-time parameters were obtained: ∆*t*1—the time of ascending front formation of *X*1 function at the vessel wall puncture moment; ∆*t*2—the time of descending front formation of *X*1 function at vessel wall puncture moment; ∆*t*3—the time of ascending front formation of *X*1 function at the moment of needle movement in surrounding tissues near the venous vessel; *x*1—the value of the integration function at the moment when the needle-electrode touches the wall of the venous vessel; *x*2—the value of the integration function at the vessel wall puncture moment; *x*3—the value of the integration function at the moment when the needle penetrates the lumen and stops; maxPr1—the maximum value of *X*1 function at the first moment of vessel puncture; Pr1 (1)—the value of *X*1 function at the first moment of vessel puncture; Pr1 (2)—the value of *X*1 function at the end moment of vessel puncture; Pr1 (3)—the value of *X*1 function at the moment of needle movement into soft tissues near the venous vessel; maxPr2—the maximum value of the derivative function at the puncture moment; minPr2—the minimum value of the derivative function at the puncture moment.

As result of the contour analysis for fifteen experimental signals, sixty-five events were received and analyzed. Among these events, fifteen events were associated with venous vessel punctures and fifty events were associated with artifacts of the needle electrode movement. Correlation analysis was performed in order to minimize the space of significant parameters. As a result, five important parameters were left. According to the results of the performed analysis, an algorithm based on logistic regression was proposed. The algorithm can allow the peak detection that corresponds to puncture. Equation (7) represents the logistic regression model [36]. To apply a logistic model, it must be adapted to the tasks being analyzed. In order to achieve this task, the regression coefficient *x* should be obtained, hence the regression coefficients were calculated using optimization techniques and gradient descent method, as shown in Equation (8).
(7)$$f\left(x\right)=\frac{1}{1+e^{-x}}$$
(8)$$X=Z_{0}+A_{\Delta t1}Z1+A_{\Delta t1}Z2+A_{\Delta t1}Z3+A_{maxPr1}Z4+A_{x2}Z5$$
where *x*—logistic function parameter; *Z*_0_—absolute term; *Z*—regression coefficient; *A*—independent variable value.

STATISTICA 10 [37] was used to implement the calculation of the regression coefficient. The regression model was trained firstly with the training sample previously obtained, which comprises fifteen experimental signals and, subsequently, sixty-five analyzed events. The puncture event was assigned a value of one, while the artifact is zero. As a result, fifteen punctures and fifty artifacts were obtained. Table 4 illustrates the result of the regression coefficients calculation.

Figure 9 shows the results of training validation; it was considered that eleven punctures and forty-nine artifacts were correctly detected.

To verify the logistic model, a control sample was collected, which consisted of nine experimental recordings and was obtained from five human subjects. For the nine experimental signals, thirty-four events were extracted, namely nine punctures and twenty-five artifacts.

The amplitude-time characteristics of the recording bio-impedance are illustrated in Table 5.

To verify the model, a contour analysis was carried out for the thirty-four events. The parameters of the regression function were found taking into account the logistic regression coefficients obtained during training. During the verification of the control samples, eight of nine punctures and twenty-four of twenty-five artifacts were correctly detected. The results of model verification on the control samples are shown in Figure 10.

Model verification showed acceptable results in terms of using the logistic regression model as the decision rule. Based on the results obtained, the sensitivity, specificity, and accuracy were 88%, 100%, and 97%, respectively.

## 5. Conclusions

A novel method for guiding the needle insertion based on electrical impedance measurement was proposed in this study. During this study, it was concluded that, in the bipolar scheme, which consists of two electrodes, one of them is the needle itself and can be used for guiding traditional needle insertion. The numerical modelling conducted to determine the appropriate location of the attached electrode regarding the vein position showed that the electrode should cover the selected vein for venipuncture in order to ensure uniform current distribution through the vein, and hence increase the sensitivity of puncture identification. However, the contact area of the attached electrode should be at least 20 × 20 mm. The maximum relative change in electrical impedance owing to a puncture of the venous vessel was achieved when the needle was removed from the attached electrode of not more than 20 mm. This arrangement of the electrode system was chosen because the relative change in electrical impedance to the first puncture of the venous vessel wall is 62%, which can be identified with high precision.

The results of theoretical studies on the influence of anatomical structures in the study area, which have different electrical resistivity, showed that, at the moment of venous vessel puncture, a jump occurs, indicating a transition from a less conductive medium to a more conductive medium. This result indicated the adequate selection of the current parameters such as the frequency and the amplitude. As shown by the results of the experimental studies, the amplitude-time characteristics of the event associated with venipuncture were determined; these parameters can be used for the technical specification of the bio-sensor such as the dynamic range and the sampling rate. Through the detailed analysis of the processes affecting the electrical impedance change, it was concluded that taking into account the influence of the needle-electrode penetration into soft tissues could help to increase the signal-to-noise ratio by an average of 24 dB, which helps to determine the moment of the first puncture of the blood vessel wall. In the course of the contour analysis, biomechanical processes of interaction between the needle electrode and the venous vessel were revealed during the first puncture of its wall.

A conclusion was made about the effectiveness of using the logistic regression model as a decisive rule for distinguishing the first puncture of the venous vessel wall from artifacts associated with the uneven speed of movement of the needle-electrode in the patient’s soft tissues. In this case, the accuracy of determining the puncture is more than 90%, which is acceptable in clinical practice, for example, to patients with poorly contoured or moving veins. This method is a prerequisite to the development of a robot-assisted venipuncture system.

## Figures and Tables

**Figure 1 sensors-22-00665-f001:**
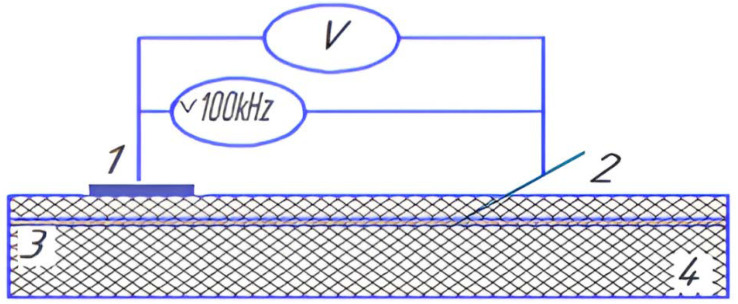
Schematic setting of the electrode system. 1, 2—electrode system, 3—blood vessel, and 4—the surrounding tissues.

**Figure 2 sensors-22-00665-f002:**
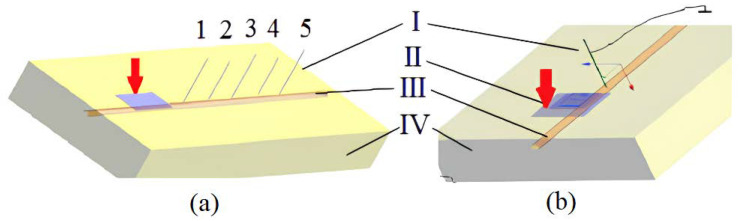
The location change (**a**) and size (**b**) of the electrode system (I—needle-electrode; II—the attached electrode; III—venous vessel; IV—soft tissue).

**Figure 3 sensors-22-00665-f003:**
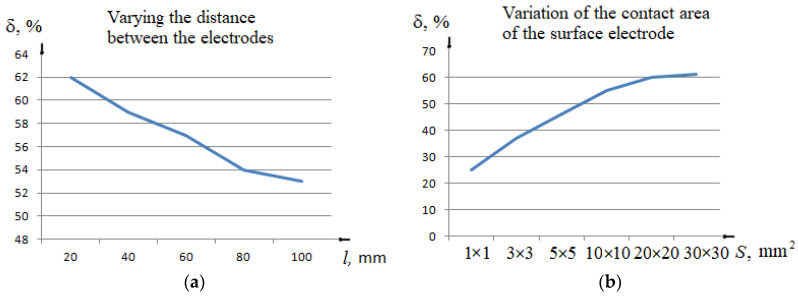
The result of sensitivity analysis of electrical impedance change due to: (**a**) variation of the distance between electrodes, (**b**) the size change of the attached electrode.

**Figure 4 sensors-22-00665-f004:**
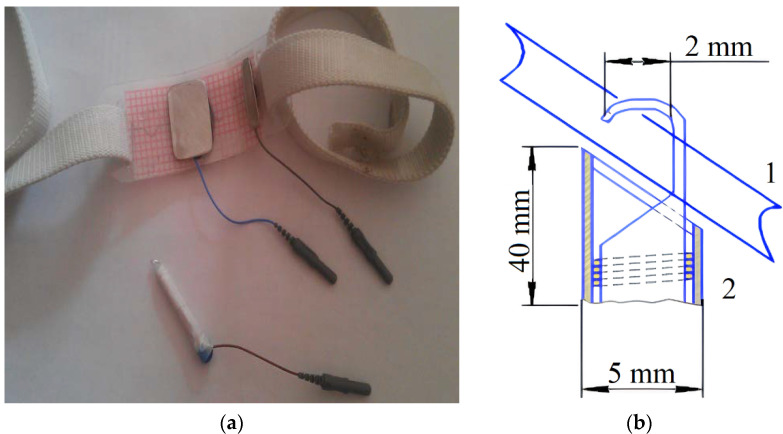
This designed electrode system: (**a**) the overview of the attached electrode with the clamp and (**b**) the clamp setup. 1—needle, 2—clamp.

**Figure 5 sensors-22-00665-f005:**
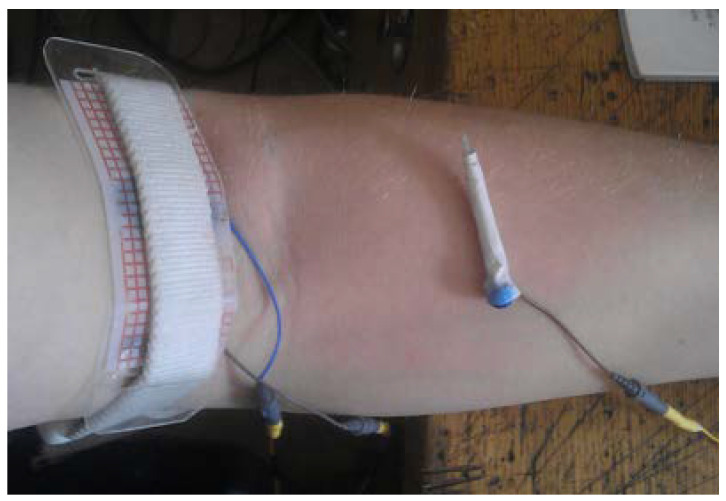
The experimental setup.

**Figure 6 sensors-22-00665-f006:**
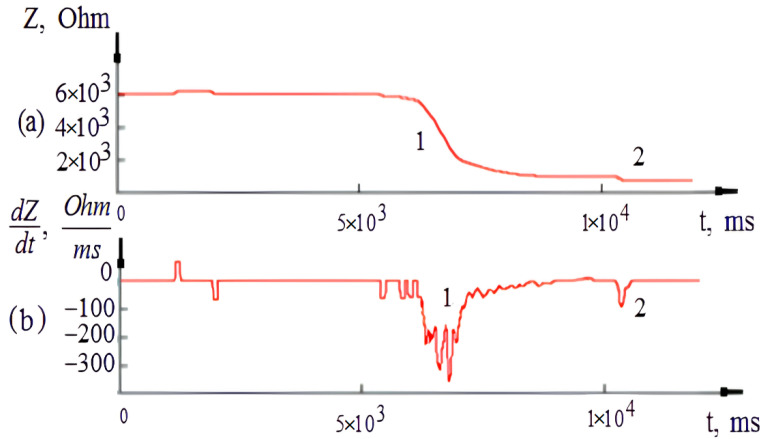
The time analysis of the recording electrical impedance signal. (**a**) The recording electrical impedance signal and (**b**) the first derivative of signal. 1—the penetration of the needle-electrode into soft tissues, 2—the first puncture of the vein wall.

**Figure 7 sensors-22-00665-f007:**
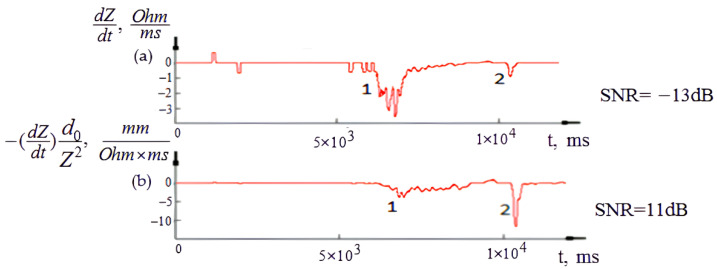
The recording electrical impedance signal and its derivative after the normalization procedure.

**Figure 8 sensors-22-00665-f008:**
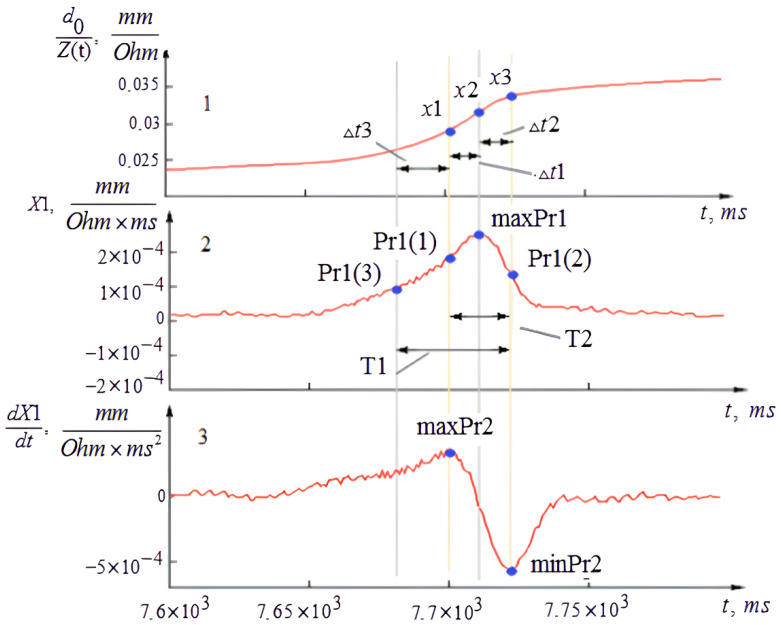
The amplitude-time characteristics of *X*1 function. 1—the integral of *X*1 function, 2—*X*1 function, 3—the first derivative of *X*1 function.

**Figure 9 sensors-22-00665-f009:**
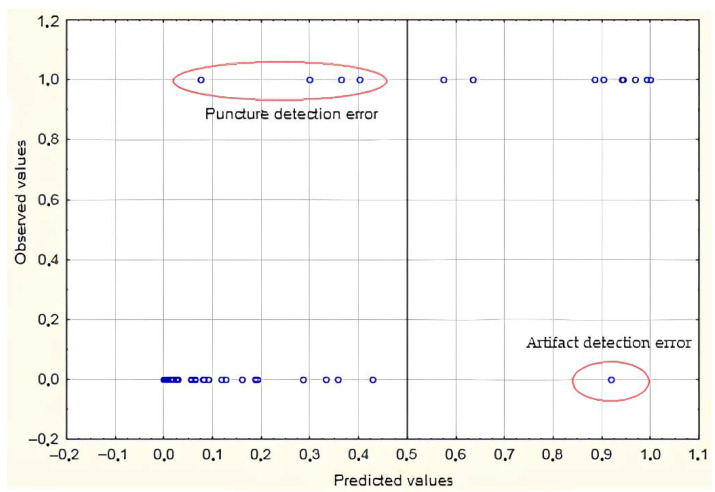
Training a logistic regression model on a test sample.

**Figure 10 sensors-22-00665-f010:**
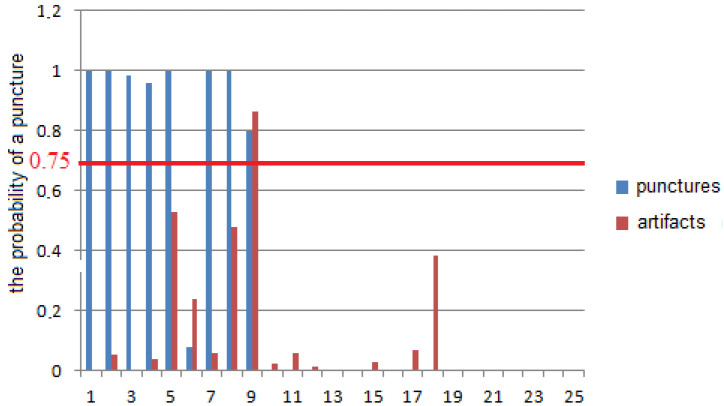
The result of model approbation on the control sample.

**Table 1 sensors-22-00665-t001:** The electrical resistivity of forearm biological tissues at 100 kHz frequency.

The Biological Tissue	The Electrical Resistivity Value, Ohm·m
Subcutaneous fat	25 ± 0.7
Nerve	12.5 ± 0.5
Blood vessel wall	3.13 ± 0.2
Muscle tissue	2.76 ± 0.3
Connective tissue	2.5 ± 0.5
Blood	1.42 ± 0.6

**Table 2 sensors-22-00665-t002:** Technical specification of the measuring system ReoCardioMonitor.

The Technical Parameter	The Value
The current	2.8 mA ± 20%
The frequency	100 kHz ± 0.5%
The impedance range	1 ÷ 240 Ohm
Sampling frequency	500 Hz
Accuracy	±0.2 Ohm
Channels number	2

**Table 3 sensors-22-00665-t003:** Characteristics of electrical impedance recording during venipuncture.

Subjects	The Number of Measurements	ΔZcommon, Ohm	ΔZpuncture, Ohm	Tpuncture Sec
1	5	5274	63	0.03
2	4	6470	139	0.04
3	2	8120	59	0.045
4	2	6958	28	0.04
5	2	7707	24	0.04

**Table 4 sensors-22-00665-t004:** The calculated regression coefficients.

The Regression Coefficients	The Value
absolute term (*z*_0_)	−0.47
∆*t*1 (*z*1)	−167.61
∆*t*2 (*z*2)	−77.21
∆*t*3 (*z*3)	−77.95
maxPr1 (*z*4)	3.82
*x*2 (*z*5)	63.94

**Table 5 sensors-22-00665-t005:** Characteristics of electrical impedance for the control group.

Subjects	The Number of Measurements	ΔZcommon, Ohm	ΔZpuncture, Ohm	Tpuncture, Sec
1	4	7220	338	0.035
2	2	6945	414	0.041
3	2	7270	317	0.04
4	1	7134	100	0.04

## Data Availability

The data presented in this study are available on request from the corresponding author.

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
