# Peer review of "Smart Bio-Impedance-Based Sensor for Guiding Standard Needle Insertion"

_sensors, 2022, doi:10.3390/s22020665_

Round 1
Reviewer 1 Report
The topic is interesting, but there are some details to be improved. The references are fine. The demonstration in the result section is not obvious.
The measurement equipment for bioZ analysis should be more described (it only measures the real part of Z i.e. the modulus, at one unique frequency of 100kHz).
The procedure is not so simple and needs clarifications on several points listed below as examples.
- In Table 1, the value for resistivity of tissues gives not common order of magnitude (and the frequency should be added).
- on simulation of the electromagnetic field and current distribution, the Fig 2 is not so clear (element II attached electrode?)
- the research ethics was followed: the reference of the ethical agreement should be given
- better explain the link between eq 3 and eq 6 to detail the normalisation procedure chosen
- why curve for X1 on Fig 8 is positive although the similar parameter on fig 7 was negative?
- for these results ("Among these events, fifteen events associated with venous vessel punctures and fifty events associated with artifacts of the needle electrode movement"), how did you discriminate?
- in the conclusion, it is said: "The conducted numerical modelling to determine the appropriate location of the attached electrode regarding the vein position showed that the electrode should cover the selected vein for venipuncture in order to make sure of uniform current distribution throw the vein and hence increase the sensitivity of puncture identification", but it is not really demonstrated in the paper (or we missed something in the explanaitions).
- maybe also justify the choice of the 100khz frequency for this study
Author Response
Dear Reviewer
Thank you very much for your time and attention to assess our manuscript, please see the attachment
with respect
Dr.Ivan

Reviewer 2 Report
Even though the manuscript can have merit, it is poorly organized and presented.
Introduction does not provide the sufficient level of background to a reader to understand why the problem of needle tip position is relevant. What is the field in which the proposed approach provides interesting advancements and what are the technological counterparts actually used.
Figure 1 is poorly prepared and described. Table 1 probably "," should be replaced by "."
In section 2.1 it is not clear what kind of numerical modelling is presented.
Table 2 - "," should be replaced by "."
I was not able to find information about ReoCardioMonitoring. More information should be provided.
Authors provided a set of equation that could be assessed during needle tip positioning. However, a validation of the proposed approach should be provided. Single analysis on a single subject is a limiting factor.
How the depth of 3.5 and 7.5 mm were evaluated to set Zmin and Zmax.? i was not able to find related information.
Figure 7 "SNR=-13 dB" please check. How SNR was evaluated?
FIgure 8 "," should be replaced by "." THe figure seems to show a "typical " change of X1. Not clear why it should be typical .
Figure 9 presented a sort of validation of the model. Why a logistic model was selected? If i have correclty interpreted the data even if theobserved value was 0 the model predicted a value ranging from 0.0 up to about 0.43. Is it correct? If yes, what are the information retrieved from this analysis.?
Table 5 provides electrical characteristics of the contro group. How they can be associated to expected or unexpected results? Reference data for comparison could be of help
Figure 10. "," should be replaced by "."
Author Response

(The authors gave the same response as above.)

Round 2
Reviewer 1 Report
Thanks to the author for replying to most of the remarks.
Maybe response 6 should be added in the text for better understanding of the curves.
Response 6: Actually in Figure 8 the signal was inverted and multiplied by (–) for simplification as well as to get rid of the minus before the function in Figure 7(b)
In addition legend of Figure 3. This designed electrode system: (a) the attached electrode; (b) clamp. could be more clear -> relative change in electrical impedance in % for 2 configurations a) and b)
Comment proposed on the electrode placement should also be added close to figure 4: However, the electrode system has an attached to the skin electrode as shown in figure 4(a) and should be tited over the arm as shown in figure 5 , this electrode has a dimension greather than the vessel diameter and should be placed over the vein from where the blood is drawn while the second electode is the clamp which shuold be connected to the needle to make it work as an electode .
The other corrections are fine.
Author Response
Dear Reviewer
Thank you very much for your attention and your time to assess our manuscript. We corrected the last remarks. Actually, in this work we have only considered one electrode system configuration but it consists of two parts that is why for more simplification and in order not to make the readers be confused , we added the paragraph written below and placed between line 197-202 above Figure 4 , as well as Figure 4(b) was changed
"However, the electrode system based on bio polar setup, the same pair of electrodes are used both for excitation and measurement. Thus, the electrode system has an attached to the skin electrode shown in Figure 4(a) with a dimension greater than the vessel diameter and should be placed over the vein from where the blood is drawn while the second electrode is the clamp which should be connected to the needle to make it work as an electrode as shown in Figure 4(b). "
With respect
Dr.Ivan
Reviewer 2 Report
Authors have sufficiently addressed the comments raised
Author Response
Dear Reviewer
Thank you very much for your attention and your time to assess our manuscript.
With respect
Dr.Ivan